# First Evidence of Antibodies Against Lloviu Virus in Schreiber’s Bent-Winged Insectivorous Bats Demonstrate a Wide Circulation of the Virus in Spain

**DOI:** 10.3390/v11040360

**Published:** 2019-04-19

**Authors:** Eva Ramírez de Arellano, Mariano Sanchez-Lockhart, Maria J. Perteguer, Maggie Bartlett, Marta Ortiz, Pamela Campioli, Ana Hernández, Jeanette Gonzalez, Karla Garcia, Manolo Ramos, Miguel Ángel Jiménez-Clavero, Antonio Tenorio, Mª Paz Sánchez-Seco, Félix González, Juan Emilio Echevarría, Gustavo Palacios, Anabel Negredo

**Affiliations:** 1Arbovirus and Imported Viral Diseases Laboratory, National Centre of Microbiology, Majadahonda, 28220 Madrid, Spain; erarellano75@gmail.com (E.R.d.A.); antonio.tenorio.matanzo@gmail.com (A.T.); paz.sanchez@isciii.es (M.P.S.-S.); 2Centre for Genome Sciences, U.S. Army Medical Research Institute of Infectious Diseases, USAMRIID, Fort Detrick, Frederick, MD 21702, USA; mariano.sanchez-lockhart.ctr@mail.mil (M.S.-L.); Maggie.l.bartlett.ctr@mail.mil (M.B.); jcgonzalezsantos@gmail.com (J.G.); Karla.y.garcia2.ctr@mail.mil (K.G.); gustavo.f.palacios.ctr@mail.mil (G.P.); 3Department of Pathology & Microbiology, University of Nebraska Medical Centre, Omaha, NE 68198, USA; 4National Centre of Microbiology, Parasitology Section, Majadahonda, 28220 Madrid, Spain; chus.perteguer@isciii.es (M.J.P.); plcampioli@isciii.es (P.C.); ahergon@isciii.es (A.H.); 5National Centre of Microbiology, Institute of Health Carlos III, Majadahonda, 28220 Madrid, Spain; mortiz@isciii.es; 6National Centre of Microbiology, Viral Immunology Laboratory, Majadahonda, 28220 Madrid, Spain; mbuylla@cbm.csic.es; 7Centro de Investigación en Sanidad Animal-Instituto Nacional de Investigación y Tecnología Agraria y Alimentaria (INIA-CISA), 28040 Madrid, Spain; majimenez@inia.es; 8Centro de Investigación Biomédica en Red de Epidemiologia y Salud Pública (CIBERESP), 28029 Madrid, Spain; jeecheva@isciii.es; 9Museo de la Naturaleza de Cantabria, Carrejo, 39592 Cantabria, Spain; fei2@ctv.es; 10National Centre of Microbiology, Virology Section, Majadahonda, 28220 Madrid, Spain

**Keywords:** Lloviu virus, Prevalence, Serology, Human, Bats

## Abstract

Although Lloviu virus (LLOV) was discovered in the carcasses of insectivorous Schreiber’s Bent-winged bats in the caves of Northern Spain in 2002, its infectivity and pathogenicity remain unclear. We examined the seroprevalence of LLOV in potentially exposed Schreiber’s Bent-winged bats (*n* = 60), common serotine bats (*n* = 10) as controls, and humans (*n* = 22) using an immunoblot assay. We found antibodies against LLOV GP_2_ in all of Schreiber’s Bent-winged bats serum pools, but not in any of the common serotine bats and human pools tested. To confirm this seroreactivity, 52 serums were individually tested using Domain Programmable Arrays (DPA), a phage display based-system serology technique for profiling filovirus epitopes. A serological signature against different LLOV proteins was obtained in 19/52 samples tested (36.5%). The immunodominant response was in the majority specific to LLOV-unique epitopes, confirming that the serological response detected was to LLOV. To our knowledge, this is the first serological evidence of LLOV exposure in live captured Schreiber’s Bent-winged bats, dissociating LLOV circulation as the cause of the previously reported die-offs.

## 1. Introduction

The *Filoviridae* family contains non-segmented RNA viruses that can cause severe haemorrhagic fever in some primates. This family is composed of five genera, *Ebolavirus*, *Marburgvirus*, *Cuevavirus, Striavirus and Thamnovirus*. The only member of the *Cuevavirus* genus discovered to date, Lloviu virus (LLOV), was described in 2011 [1,2,3]. LLOV is believed to be the first filovirus detected in Europe that was not imported from an endemic area in Africa or Asia. LLOV RNA was found in the lung, liver, rectal swab, and/or spleen of several Schreiber’s Bent-winged bats’ carcasses in 2002 [1]. Since then, hundreds of oral and rectal swabs of live captured Schreiber’s Bent-winged bats from Spain were screened during 2002 to 2009, and no LLOV RNA was detected. Moreover, other bat species sampled in the same caves where LLOV was originally detected were also negative for LLOV RNA [1]. In contrast, fresh carcases of Schreiber’s Bent-winged bats recovered in 2016 from Northeastern Hungary (Bükk Mountain) were positive for LLOV RNA, demonstrating that LLOV was still circulating in Europe [4].

Bats have been implicated as reservoirs of filoviruses in Africa and Asia after specific antibodies and nucleic acids were detected in fruit and insectivorous bats [5,6,7,8,9,10,11,12,13,14]. Marburg virus (MARV) was isolated from wild-caught Egyptian rousette bats’ tissues [15,16]. Recently, Towner et al. demonstrated MARV transmission from inoculated to naïve Egyptian rousette bats [17], establishing Egyptian rousette bats as a natural reservoir of *Marburgvirus* (MARV and Ravn virus, RAVV). A seroprevalence of 20.5% was established in wild-caught Egyptian rousette bats from the Democratic Republic of the Congo [7]; 43.8% from Zambia [18] and 14.8% and 21.5% from juvenile and adult bats, respectively, captured in the Python Cave in Uganda [16]. In addition, the complete genome of Bombali virus (BOMV), a novel genera, was detected in the faeces of little free-tailed bats (*Chaerephon pumilus*) and Angolan free-tailed bats (*Mops condylurus*), demonstrating that bats, at a minimum, are part of the filovirus transmission cycle [14]. More recently, in March 2019, another novel filovirus, “Měnglà virus”, has been isolated from *Rousettus* bats in China [19].

Previous to this study, LLOV had only been detected after Schreiber’s Bent-winged bats’ die-offs. This is relevant in the debate regarding the filovirus reservoir, since the current paradigm associates reservoirs with low virulence [20] or tolerance [21]. In that context, the relation between LLOV and die-offs is a rarity. Thus, the capacity of LLOV to infect animal species different from Schreiber’s Bent-winged bats, and its potential to cause disease in bats and humans, remains a puzzle.

The biological properties of LLOV remain mostly uncharacterized, since infectious LLOV has not been isolated yet. LLOV has a genomic organization similar to those of *Ebolavirus* and *Marburgvirus* members, with a single-stranded, negative-sense RNA genome, 19 kb in length, that contains 7 open reading frames (ORF), encoding for the nucleoprotein (NP), viral protein-35 (VP35), VP40, glycoprotein (GP), VP30, VP24, and RNA-dependent RNA polymerase (L) proteins. The expression of recombinant LLOV GP had been used to investigate its structural and functional properties. LLOV GP is responsible for both receptor binding and fusion of the virus envelope with the host cell membrane [22,23,24,25,26,27,28,29,30]. Filovirus GP undergoes proteolytic cleavage by host proteases such as furin, resulting in two subunits, GP_1_ and GP_2_, which are linked by a disulphide bond [26]. GP is highly N- and O-glycosylated in its middle section, which is thus designated the mucin-like region. Several reports had demonstrated GP antigenicity making it the target of choice for serological studies that estimate exposure and prevalence [27,28,29,30].

Along those lines, we collected serum from wild-caught Schreiber’s Bent-winged bats and common serotine bats (*Eptesicus serotinus*), to be used as negative controls, and from humans with a history of exposure to bats to establish LLOV seroprevalence. Our study demonstrates LLOV exposure in live captured Schreiber’s Bent-winged bats, but not in common serotine bats or humans. Our data confirms that Schreiber’s Bent-winged bats were exposed to LLOV at the two caves where it was originally discovered. We discuss the significance of this finding regarding the pathogenicity of LLOV, and compare it to available MARV seroprevalence data in Egyptian rousette bats.

## 2. Materials and Methods

### 2.1. Human and Bat Serum Samples

Characteristics and origin of the sera used in the study are summarized in Table 1. Group 1 (pools H1-H4) consisted of 22 human sera from bat handlers with a known history of exposure to Schreibers’ bats from different Spanish caves during 2003 to 2008. Human samples were collected under the protocol of project SAF2009-09172, approved by the General Research Programme of the Spanish Government on 20 November 2009. Group 2 was comprised of 60 sera from Schreiber’s Bent-winged bats, live captured in 2015 from the Asturias (pools A1–A5) and Cantabria (pools C1-C7) caves in the North of Spain, where the Lloviu Virus (LLOV) was originally detected in deceased Schreiber’s Bent-winged bats in 2002. Group 3 (pools E1 and E2) included 10 common serotine sera bats live captured in different locations in Huelva, Andalusia, South of Spain, used as negative controls. All samples were analysed in pools containing 2–5 sera per pool (Table 1). Sample collection in Cantabria was approved by “Consejería de Ganadería, Pesca y Desarrollo Rural” of the Government of Cantabria’’ (EST 702/15 SEP). Bat samples’ collecting was approved in Asturias by “Consejería de Agroganadería y Recursos Autóctonos” of the Government of the Principality of Asturias (dossiers 2015/007804). Bats were captured with mist-nets near roots, and released at the same collection point after being identified, measured, sex determined, and sampled. A venous blood sample was collected as described in Smith et al. [31] and the supernatant was recovered after 24 h and maintained at -20 ºC, until it could be sent to the Spanish National Centre of Microbiology. The serum samples obtained were not inactivated.

Stool samples of 40 Schreiber’s Bent-winged bats captured in Asturias and 40 Schreiber’s Bent-winged bats captured in Cantabria in 2015 were collected and homogenized in 1 mL of lysis buffer. Samples were maintained at -20 ºC for 48 h before transport to the Spanish National Centre of Microbiology, where they were stored at -80 ºC until processing.

### 2.2. Recombinant LLOV GP Antigen

The C-terminal domain of LLOV glycoprotein (GP) (GP_2,_ GenBank AN: JF828358) was utilised as an antigen for the immunoblot assays and CAT (Chloramphenicol Acetyl Transferase) as the control protein (Figure 1a,b). The 963 bp amplified GP_2_ fragment was directionally subcloned into a pfastBac HT B donor plasmid (Invitrogen, cat nº10359-016) with a hexahistidine (6× His) tag sequence before transposition into a bacmid (Invitrogen, cat nº10359-016) for protein production. Purified recombinant Bacmid-6×His-LLOV-GP_2_ viral stocks were used to infect a *Spodoptera frugiperda* 21 (Sf21) insect cell line (5 × 10^5^ cells/mL). The 40 kDa recombinant 6xHis-LLOV-GP_2_ protein used as the antigen was obtained from a crude extract of the pellet fraction after treatment with Inclusion Body Solubilisation reagent (IBS, Thermo Fisher scientific). A detailed summary of the antigen production process is included in a supplementary text (see Appendix A).

### 2.3. Detection of LLOV RNA by Real Time PCR

Viral RNA was extracted from the stool of 80 Schreiber’s Bent-winged bats captured in Cantabria in 2015, using the QIAamp RNA Viral Kit (Qiagen GmbH, Heiden, Germany) according to the manufacturer’s recommendations. A qRT-PCR to detect LLOV was performed as described by Negredo et al. [1], using modified primers and a probe for LLOV detection from the method described by Panning et al. [32]. We used primers pair: FiloAneo: 5´-ARG CMT TYC CAN GYA AYA TGA TGG T-3´ and FiloBNeo: 5´-RTG WGG NGG RYT RTA AWA RTC ACT NAC ATG-3´ and the probe Lloviu-S: FAM-5´-CCT AGA TTG CCC TGT TCA TGA TGC CA-BHQ1-3´. Briefly, cDNA was synthesised in the presence of an RNase inhibitor (Invitrogen) using the SuperScriptTM III Reverse Transciptase kit (ThermoFisher, Spain) following manufacturer instructions. The qPCR was carried out using a commercial kit (LightCycler® TaqMan, Roche, Mannhenn, Germany). For the assay, 5 µL of sample cDNA was mixed with a 15 µL reaction mix. Amplification conditions consisted of an initial DNA denaturalisation of 10 min at 95 °C, and 45 cycles of 15 s at 95 °C, and 1 min at 60 °C for annealing, and 72 °C for extension. qPCR was carried out in a Roche LightCycler® 2.0. Fluorescence was measured during the 60 °C step. We also included an internal competitive control to detect false negative results. It consisted of a DNA insert of 75 bp obtained with the primer pair Upper 5´-AGG CAT TCC CGA GCA ACA TGA TGG TCC AGC ACA CAT GTG TCT ACT-3´ and Lower 5´-GTG AGG GGG GCT GTA ATA GTC ACT GAC ATG AGT AGA CAC ATG TGT GCT-GG-3´, cloned in a pCR4-TOPO TA cloning vector (TOPO TA Cloning Systems, Invitrogen, Spain).

### 2.4. Detection of LLOV-Specific Antibodies by Immunoblot

Prior to use, the recombinant 6xHis-LLOV-GP_2_ protein was tested for reactivity in immunoblots to an anti-LLOV GP polyclonal mouse serum (kindly provided by Dr A. Takada from the Research Centre for Zoonosis Control of Hokkaido University) [22]. Human samples were grouped into 6 pools with a maximum of 5 sera from the same year (Table 1). Bat samples were grouped in 14 pools with a maximum of 5 sera each, according to their species and location of origin. Due to the limited amount of antigen obtained, only 10 sera from 2 positive pools from Cantabria were analysed individually (Table 1).

Briefly, the immunoblot LLOV procedure was conducted as follows: 10 µl of the recombinant GP_2_ crude extract was used as the antigen, loaded in single-well SDS-PAGE 12.5% gels, run and electro-transferred to nitrocellulose membranes, following standard procedures. Human sera (10 µL) were diluted 1:50, and detected with an anti-human polyclonal antibody conjugated with peroxidase diluted 1:4000 (anti-human IgG Fc-HRP, cat. 9040-05, southern Biotech). Bat sera (5 µL) were diluted 1:100 and detected with a peroxidase conjugated goat anti-bat IgG (cat. A140-118P, Bethyl). Wells were blocked with non-fat milk powder (2.5% solution) and the colour reaction was developed using the CN/DAB substrate kit, (Thermo scientific, cat. 34000).

An anti-LLOV GP polyclonal mouse serum provided by Dr A. Takada was used as the positive control. The antibody was diluted 1:2000, and detected with an anti-mouse polyclonal IgG HRP-link as secondary antibody (cat. 7076; Cell Signaling Technology). CAT crude extracted was included in each assay as the negative control and was detected using the same conditions and conjugates for each species.

### 2.5. Detection of LLOV-Specific Antibodies by Domain Programmable Arrays

Sera from 52 wild-caught Schreiber’s Bent-winged bats were analysed using DPA, as described [33]. A phage library displaying peptides from all open reading frames (ORFs) of every published filovirus [Ebola (EBOV), Sudan (SUDV), Taï Forest (TAFV), Bundibugyo (BDBV), Reston (RESTV), Marburg (MARV), Ravn (RAVV) and LLOV], was used for the serological assessment of the humoral response. The phage library was generated using 36 amino acid oligonucleotides designed to tile all ORFs every 7 amino acids. IgG antibodies from individual Schreiber’s Bent-winged bats sera were captured in the solid phase using goat anti-Bat IgG (Novus Biologicals, Catalogue number NB7237, 1 mg/mL). All samples were processed in triplicates. As a positive control, a hyperimmune polyclonal serum against EBOV was used. The original filovirus phage library, or the recovered phages after panning (bound fraction), were lysed, cloned, and PCR-amplified [33]. Triplicates were index coded and pooled, before a dual index Illumina preparation was performed using the Apollo 324 robot [Wafergen]. Individual libraries were pooled together at a final concentration of 2nM, and sequenced using a MiSeq DNA sequencer instrument employing a 600 cycle kit (2 × 250 cycles) with 20% of PhiX. A minimum target sequence depth of approximately 500 K reads per library was pursued. The process of identifying clusters of enriched displayed peptides after panning (using their encoding oligonucleotide information) has been described [33]. Sequencing reads were then run through the in-house bioinformatics pipeline that includes: Array Description, Input Randomisation, Removal of Duplicates and Read Mate Correction, Read Cleaning, and Expression Analysis. The pipeline for DPA analysis can be downloaded from github: https://github.com/kygarcia/DPA_Analysis_Pipeline [33]. The oligo clusters identified in multiple individuals were separated into 8 amino acid peptides, ordered with long-chain biotin attached after a glycine linker to the N-terminal end, and tested via western blot to identify immunodominant epitopes. EXP is a value calculated based on cumulative counts and expression, to provide an adjusted counts and expression for each oligonucleotide

### 2.6. Protein Modelling and Surface Epitope Visualization

The crystal structures of EBOV proteins have been defined for some strains, but not all proteins of interest existed within one EBOV strain. Further, LLOV protein crystal structures have not been created. For consistency the amino acids of each protein (EBOV GP: APT69657.1, EBOV NP: AXE75587.1, EBOV VP35: AXE75588.1, LLOV GP1: YP_004928138.1, LLOV GP2: YP_004928139.1, LLOV NP: YP_004928135.1, LLOV VP35: AER23672.1) were used to generate a probable monomer model for each protein using Phyre2 to predict folding [34]. Models were overlaid and labelled to show any surface exposure of epitopes using PyMOL Molecular Graphics System (version 2.0, Schrödinger, LLC).

## 3. Results

### 3.1. Absence of LLOV RNA in the Schreiber’s Bent-Winged Bats

No LLOV RNA was detected in any of the stool samples from Schreiber’s Bent-winged bats captured in 2015 in the same caves where LLOV was found in 2002 [1]. Also, in contrast with the Schreiber’s Bent-winged bats from 2002, all Schreiber’s Bent-winged bats surveyed in 2015 presented no overt signs of disease. 

The inability to find LLOV RNA in the Schreiber’s Bent-winged bats stool samples that were analysed, could indicate that either LLOV does not normally circulate in this bat species, and the die-off episode was an oddity, or that the prevalence of active LLOV infection, with viremia and virus shedding, is normally low in Schreiber’s Bent-winged bats, requiring a higher number of surveyed animals to increase the chances of detection. However, to prove this likelihood, further analysis of other tissues from the same bat population is required.

### 3.2. No Detectable Antibodies against LLOV GP_2_ Protein Were Found in the Human Samples by Immunoblot

Since the glycoprotein (GP) of filoviruses has been shown to be one of the main antigens during infection [27,28,29,30], we developed an immunoblot assay to detect specific humoral responses against LLOV GP. We attempted to express and purify both LLOV GP_1_ and GP_2_ subunits, but only GP_2_ production was successful (Figure 1b). A total of 22 human sera, collected between 2003 and 2008 from bat handlers working in Spanish caves where Schreiber’s Bent-winged bats inhabit, were analysed with this assay. None of the human sera pools analysed presented detectable reactivity against the recombinant LLOV GP_2_ subunit (LLOV-rGP_2_) (Table 1).

### 3.3. Absence of Antibodies against LLOV GP_2_ Protein in Common Serotine Bats by Immunoblot

A total of 10 sera from common serotine bats captured from caves within the province of Huelva, Andalusia, were grouped into 2 pools, and humoral response to LLOV was analysed by Immunoblot. We used the sera from unaffected bat populations with a completely different ecology from Schreiber’s Bent-winged bats, as negative controls to identify possible unspecific results. None of the pools analysed presented detectable reactivity against LLOV-rGP_2_ by Immunoblot (Table 1), indicating the absence of antibodies against this subunit, and suggesting no LLOV exposure in the common serotine bats surveyed.

### 3.4. Presence of Antibodies against LLOV GP_2_ Protein in Schreiber’s Bent-Winged Bats by Immunoblot

A total of 60 serum samples from 2015 cave-caught (Cantabria or Asturias) Schreiber’s Bent-winged bats were analysed. Strikingly, all 7 pools from the Cantabria cave, and 5 from the Asturias cave (Cueva del Lloviu) had detectable reactivity against LLOV-rGP_2_ by Immunoblot (Table 1; Figure 1c). Sera from 2 of the pools were further tested individually. Four of the sera from the C2 pool, and one serum from the C1 pool were reactive against LLOV-rGP_2_ by immunoblot. Unfortunately, the limited amount of LLOV-rGP_2_ antigen prevented further analysis of individual sera by immunoblot.

### 3.5. Presence of Antibodies against LLOV Proteins in Schreiber’s Bent-Winged Bats Were Confirmed Using DPA

To confirm and expand the results obtained by immunoblot, we evaluated the presence of specific antibodies against filoviruses using the DPA assay [33]. We tested 52 sera overall; 12 of 32 (37.5%) and 7 of 20 (35.0%) individual Schreiber’s Bent-winged bats serum from the Cantabria and Asturias caves, respectively, presented strong evidence of LLOV exposure (Table 1) against multiple LLOV proteins (Table 2). The remaining samples tested from both caves were negative by this assay. 

Although generally in agreement, some discordances were observed between immunoblot and DPA. Only 8 individual samples were analysed both by DPA and immunoblot (Table 1). Two of them were positive by immunoblot (samples 4 and 6 from Cantabria Cave) but negative by DPA. All other individual serum results were concordant. When the DPA results from individual serum was compared with the immunoblot results of the associated pools, 2 of 12 pools were also found to be discordant (Table 1). As mentioned above, the only individual positive serum by immunoblot in pool C1 (sample 4) was negative by DPA, while all five individual sera from pool A2 were negative by DPA (sample 10B, 12B, 13B, 15B and 16B). Nonetheless, to estimate the seroprevalence of LLOV in Schreiber’s Bent-winged bats, we only considered the results in which both assays correlated.

### 3.6. Detection of Significant Prevalence of LLOV in Schreiber’s Bent-Winged Bats

The resulting LLOV prevalence in Schreiber’s Bent-winged bats was 36.5% (19/52). We did not detect a significant difference from animals collected in different regions (37.5% in Cantabria; 35% in Asturias), neither in association with gender (males: 33% (9/27); females 40% (10/25). The prevalence of LLOV was slightly higher in adults (41.6%, 5/12) than in juveniles (35%, 14/40) as was also reported for MARV seroprevalence in Egyptian rousette bats in the Python cave [16] and in the Kitaka cave [15]. These reflect what we expect is the seroprevalence in this population, however, more samples are needed to avoid a batch effect bias.

### 3.7. Immunodominance Epitope Analysis in Schreiber’s Bent-Winged Bats

We previously reported the immunodominant epitopes of Ebola virus (EBOV) GP recognised by the humoral response in non-humans primates (NHPs) after vaccination [33]. Utilising the same approach, we analysed the DPA results to determine the LLOV epitope pattern of recognition in Schreiber’s Bent-winged bats. Several epitopes were detected among distinct LLOV proteins, mainly in GP, nucleoprotein (NP) and viral protein (VP) 35, and to a less significant degree in VP40, VP24 and L ORFs (Table 2). The most prevalent epitopes detected were named with the following notation: Name of the protein they belong to, followed by a number that represents the N-terminal amino acid position of the epitope (i.e., “GP2.28”, is an epitope at the GP_2_ subunit, starting at amino acid 28) (Figure 2).

## 4. Discussion

During the last few decades, an intensive search to identify filovirus reservoir hosts in wildlife populations has resulted in the detection of antibodies against Ebola and Marburg viruses in several species of bats [6,7,8,10,11,12,13]. While seroprevalence was somehow widespread, direct virus detection was scarce. 

MARV and RAVV have been isolated from Egyptian rousette bats in Africa [7,8,17], but only limited detection of filovirus RNA in certain species of African and Asian bats [5,7,8,15] has been reported. Indeed, neither EBOV nor MARV have been associated with bat mortality. In contrast, LLOV was the suspected cause of massive bat mortality in caves from northern Spain affecting a single bat species, Schreiber’s Bent-winged bats, in 2002 [1,35], but the pathogenicity of this virus in bats remains unclear. In this study we aimed to clarify this, and searched for the presence of the virus in live-captured Schreiber’s Bent-winged bats from the same caves where the LLOV was discovered. However, we do not have any data to demonstrate the bats captured in 2015 belong to the same colony that was affected in 2002. It is possible, or even likely, that the original colony had dispersed or perished and another sub-population then filled the roost site. On the other hand, despite testing hundreds of samples from Schreiber’s Bent-winged bats since 2002, including from the same caves, LLOV RNA was not detected again until 2016, when fresh carcases were recovered from Northeastern Hungary [4]. The explanation for this could be the fact that Schreiber’s Bent-winged bats populations are declining across Europe, mainly because of human disturbance. Most of their summer or winter roosting sites are unknown, or if known, they are protected habitats, and rarely reached by specialists.

In this context, serological surveys constitute a valuable resource in the search for evidence of circulation of LLOV. A GP-based enzyme-linked immunosorbent assay (ELISA) for the detection of antibodies against LLOV has been developed, demonstrating the antigenicity and specificity of the target [22,23]. Therefore, we used the C-terminal domain of the LLOV GP (LLOV-rGP_2_) as an antigen in an immunoblot assay. The assay was used to measure LLOV-specific antibodies in live-captured Schreiber’s Bent-winged bats from the same caves affected by the die-off in 2002. We further confirmed these results using an independent assay, DPA [33]. We observed some discrepancies in the results of both assays that could be due to the higher immunoblot sensitivity compared with DPA, since we are utilising a very conservative threshold in the DPA enrichment factor (Material and Methods). Combining both assays, we were able to show strong evidence of LLOV exposure in 19 of 52 Schreiber’s Bent-winged bats samples tested (36.5%). This seroprevalence level is similar to that previously described for EBOV in other bat species [6,12] and MARV in Egyptian rousette bats [7,16,18], and suggests that Schreiber’s Bent-winged bats were exposed to LLOV without associated mass-mortality. In addition, our study excluded performing neutralisation tests with the sera of bats, since a recent study demonstrated that antibody-mediated virus neutralisation does not contribute significantly to the control and clearance of Marburg virus, Ebola virus or Sosuga virus infection in Egyptian rousette bats [36].

To further demonstrate specificity, we compared the immunodominance pattern between Schreiber’s Bent-winged bats and primates, with the assumption that a bona-fide response in this bat species should overlap (at least partially) with the known response in primates. The domain that contains most of the LLOV NP epitope targets recognised by the humoral immunity of Schreiber’s Bent-winged bats (LLOV NP.511–525, NP567–581 and NP.630–638) overlaps with what was previously reported in other mammals, including humans (Figure 3). Numerous EBOV NP epitopes were described within this region (EBOV NP 521–540, 561–580, and 632–645), despite a low residue identity (22.2%–33.3%) in the area. This trend is conserved in the response against BDBV, RESTV, SUDV, TAFV or EBOV [37]. Likewise, most of the 54 epitopes described in humans against EBOV GP fall within the glycan cap [38], and most immunodominant epitopes in NHPs are directed against the glycan cap and mucin-like domain [33]. Further, epitope prediction performed using the BepiPred server (www.cbs.dtu.dk/services/BepiPred/) identified the glycan cap of EBOV, TAFV, SUDV, RESTV, and MARV as highly immunogenic and immunodominant, despite being highly divergent [38]. Two epitopes were widely detected by the Schreiber’s Bent-winged bats: GP1.224, within the GP glycan cap, which overlays with epitopes detected in EBOV GP (Figure 3); and GP2.28, which lies 3 amino acids upstream of the predicted region of antigenicity [39]. 

Interestingly, although we did not identify any immunodominant epitopes within LLOV VP40 in apparent conflict with the immunodominance of this ORF in EBOV and SUDV studies [40,41], we found several epitopes in LLOV VP35. EBOV VP35 is immunogenic in humans [41]; in fact, 83 epitopes have been identified in EBOV VP35. The LLOV epitopes VP35.189 and VP35.243 directly overlap with those seen in EBOV VP35 (Figure 3). In summary, there is a clear correlation between the areas of immunodominance of the serological response against LLOV in Schreiber’s Bent-winged bats with the response against filovirus in primates. Although we cannot completely rule out that the humoral response observed in Schreiber’s Bent-winged bats could have originated from a still unknown, closely-related *Cuevavirus*, the immunogenic correlation observed, and the fact that most of the Schreiber’s Bent-winged bats scored as “positive” for LLOV, detected several LLOV specific epitopes (Table 2), the most likely scenario is that the *Miniopterus schrebersii* bats from Spain have been exposed to LLOV.

On the other hand, the absence of RNA in faecal specimens of the same bat populations where LLOV-antibodies were detected, is reminiscent of similar findings described for EBOV and MARV [6,7,10,11,12,13], suggesting that virus exposure in younglings promotes seroconversion, and protects the animals from future LLOV infection and viral shedding. An alternative explanation would imply a very rapid host-virus adaptation, with the selection of naturally resistant individuals after the die-off. In that scenario, less plausible in our opinion, repeated exposure to LLOV in resistant animals would lead to a decrease in the circulation of the virus, with an increase in herd immunity. Both the high viral loads observed in the carcasses [1], and the presence of significant seroprevalence of anti-LLOV antibodies in live-captured Schreiber’s Bent-winged bats, appear to support both scenarios. 

In conclusion, we demonstrated the presence of anti-LLOV antibodies in live-captured Schreiber’s Bent-winged bats taken in the same caves were LLOV was originally discovered. Consequently, survival of Schreiber’s Bent-winged bats after exposure to LLOV seems to be a frequent event, suggesting that LLOV might not be highly pathogenic for Schreiber’s Bent-winged bats. Conversely, LLOV may be highly pathogenic for Schreiber’s Bent-winged bats, and thus this study may be the first evidence of immunity in the surviving animals after initial LLOV exposure. As many scenarios are possible, more extensive serosurveys, including other geographical locations, as well as other ecologically related bats (*Myotis* genus), are needed to conclusively confirm which of these occurred.

## Figures and Tables

**Figure 1 viruses-11-00360-f001:**
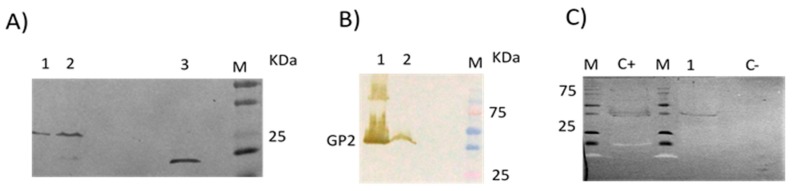
(**A**) Expression of the recombinant CAT protein (28 KDa) in the pellet (lane 1) and the supernatant (lane 2) of the crude extracted lysate by Immunoblot, revealed with anti-His antibody (His Tag Mouse mAb HRP conjugate, dilution 1:2500), and anti-mouse IgG, HRP-link antibody, as a secondary antibody (dilution 1:2000). A non-related His-recombinant protein, treated in the same conditions, was loaded as a reaction positive control (lane 3). (**B**) Expression of the recombinant Lloviu Virus (LLOV) GP_2_ protein (40 KDa) in the supernatant (lane 1) and pellet (lane 2) of the crude extracted lysate, revealed with the same antibodies and conditions. M = Molecular weight markers. (**C**) Reactivity of one of the 7 positive serum pools from *M. schreibersii* bats of Cantabria caves (C5, Table 1) (lane 1) by Immunoblot. An anti-LLOV glycoprotein (GP), polyclonal mouse serum (dilution 1:200) was used as positive control (C+). As negative control (C-), the same amount of CAT crude extract was used, revealed with the same antibodies and conditions.

**Figure 2 viruses-11-00360-f002:**
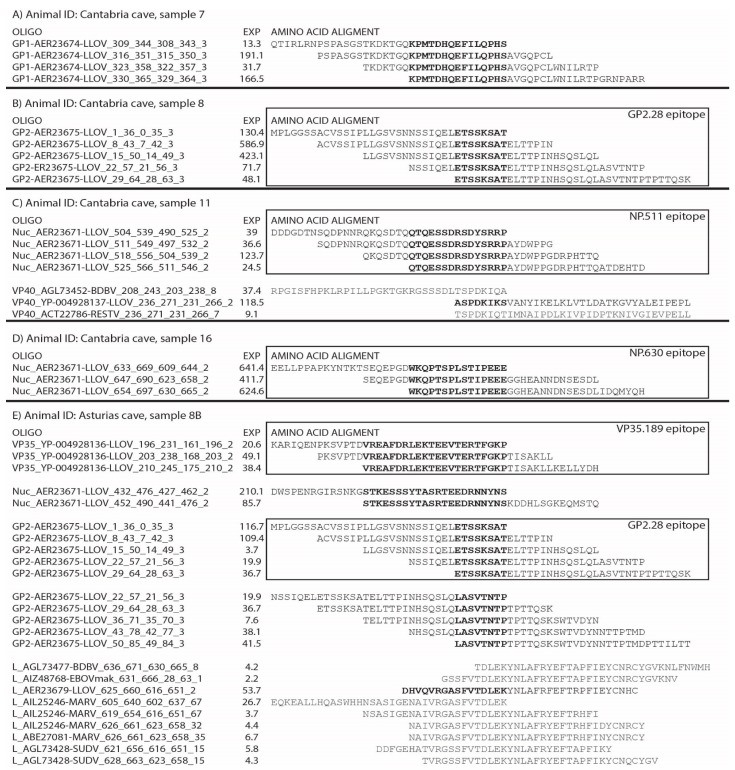
Representative examples of clusters of enrichment found in different Schreiber’s Bent-winged bats. In bold are the regions in common, within the cluster of enrichment (putative epitope). Some animals presented only one cluster, in GP_1_, GP_2_, or the nucleoprotein (NP); for animal (A, B, D) respectively. Other bats presented several clusters, in different proteins, NP and VP40, and viral protein-35 (VP35), NP, GP_2_ and L; for animal (C) and (E), respectively. In the squares are represented immunodominant epitopes, GP2.28, NP511, NP 630 and VP35.189. Note: Due to the similarity on the L protein in this region within filoviruses, the last cluster of E) is a hybrid cluster, with phages encoding proteins from LLOV, BDBV, EBOV, MARV and SUDV. “Nuc” corresponds to the nucleocapsid protein (NP), “OLIGO” to the oligonucleotides forming the cluster of enrichment.

**Figure 3 viruses-11-00360-f003:**
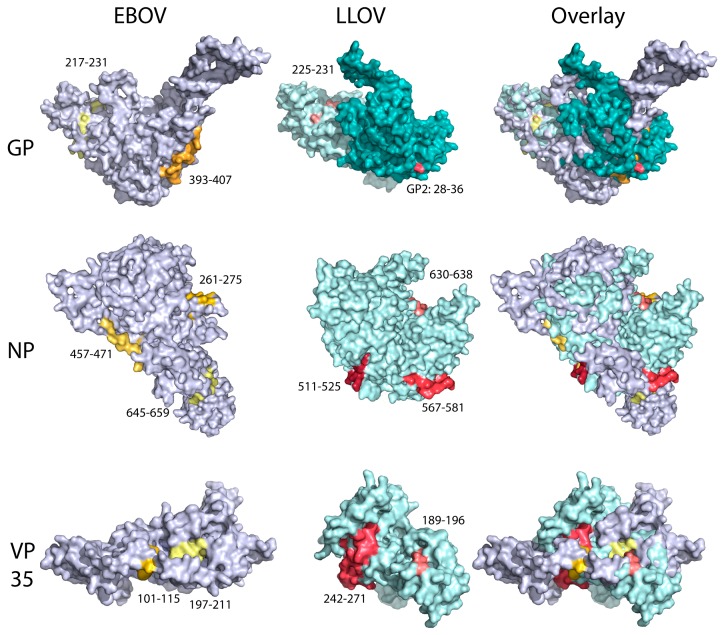
Structural modelling of Ebola (EBOV) (left), Lloviu (LLOV) (middle) and EBOV/LLOV overlay proteins (right). EBOV epitopes are in yellow, gold, and orange (GP: 217–231,393–407; NP: 261–275, 457–471, 473–487; VP35:197–211, 266–280) while LLOV epitopes described are in pink, salmon, and red (GP: 225–231, GP2 28–36, NP: 511–525, 567–581, 630–638, VP35: 189–196, 242–271). LLOV epitopes detected by Schreiber’s Bent-winged bats are in similar regions to the ones described in humans, even when the amino acid sequence of the proteins are dissimilar.

**Table 1 viruses-11-00360-t001:** Description of all sera used in the study. Samples are described according to the pool there were grouped, region of collection, sex, age range, immunoblot and Domain Programmable Arrays (DPA) results. ND: Not Determined.

Pools	Serum Number	Species	Year	Location	Gender	Animal Age	Sample Number	Pool Immunoblot	Individual Immunoblot	DPA
H1	4	Human (*H. sapiens*)	2003	Sevilla	Female	Unknown	1	Neg	ND	ND
					Male	Unknown	2		ND	ND
					Male	Unknown	3		ND	ND
					Male	Unknown	4		ND	ND
H2	4	Human (*H. sapiens*)	2004	Sevilla	Female	Unknown	5	Neg	ND	ND
					Male	Unknown	6		ND	ND
					Male	Unknown	7		ND	ND
					Male	Unknown	8		ND	ND
H3	5	Human (*H. sapiens*)	2006	Vizcaya	Male	Unknown	9	Neg	ND	ND
				Navarra	Male	Unknown	10		ND	ND
				La Rioja	Male	Unknown	11		ND	ND
				Vizcaya	Male	Unknown	12		ND	ND
				Murcia	Male	Unknown	13		ND	ND
H4	5	Human (*H. sapiens*)	2006	Badajoz	Female	Unknown	14	Neg	ND	ND
				Vizcaya	Male	Unknown	15		ND	ND
				Madrid	Male	Unknown	16		ND	ND
				Badajoz	Male	Unknown	17		ND	ND
				Vizcaya	Male	Unknown	18		ND	ND
H5	2	Human (*H. sapiens*)	2007	Barcelona	Female	Unknown	19	Neg	ND	ND
				Sevilla	Female	Unknown	20		ND	ND
H6	2	Human (*H. sapiens*)	2008	Sevilla	Male	Unknown	21	Neg	ND	ND
					Female	Unknown	22		ND	ND
E1	5	Serotine bat (*E. serotinus*)	2002	Huelva	Unknown	Unknown	1	Neg	ND	ND
					Unknown	Unknown	2		ND	ND
					Unknown	Unknown	3		ND	ND
					Unknown	Unknown	4		ND	ND
					Unknown	Unknown	5		ND	ND
E2	5	Serotine bat (*E. serotinus*)	2000	Huelva	Unknown	Unknown	6	Neg	ND	ND
					Unknown	Unknown	7		ND	ND
					Unknown	Unknown	8		ND	ND
					Unknown	Unknown	9		ND	ND
					Unknown	Unknown	10		ND	ND
C1	5	Schreibers’ bat (*M. schreibersii*)	2015	Cantabria	Male	Young	2	Pos	Neg	Neg
					Male	Young	3		Neg	Neg
					Female	Young	4		Pos	Neg
					Female	Adult	1		Neg	ND
					Female	Young	5		Neg	Neg
C2	5	Schreibers’ bat (*M. schreibersii*)	2015	Cantabria	Female	Adult	6	Pos	Pos	Neg
					Female	Adult	7		Pos	Pos
					Female	Adult	8		Pos	Pos
					Female	Young	12		Neg	ND
					Female	Young	11		Pos	Pos
C3	5	Schreibers’ bat (*M. schreibersii*)	2015	Cantabria	Male	Young	13	Pos	ND	Neg
					Male	Young	14		ND	Pos
					Female	Young	15		ND	Pos
					Female	Adult	16		ND	Pos
					Female	Adult	18		ND	Neg
C4	5	Schreibers’ bat (*M. schreibersii*)	2015	Cantabria	Female	Young	19	Pos	ND	Neg
					Male	Young	20		ND	Pos
					Female	Adult	21		ND	ND
					Male	Young	22		ND	Neg
					Male	Young	23		ND	Neg
C5	5	Schreibers’ bat (*M. schreibersii*)	2015	Cantabria	Female	Young	24	Pos	ND	Neg
					Female	Adult	25		ND	Neg
					Female	Young	26		ND	Neg
					Male	Young	27		ND	Pos
					Female	Young	32		ND	Neg
C6	5	Schreibers’ bat (*M. schreibersii*)	2015	Cantabria	Male	Young	33	Pos	ND	Neg
					Male	Young	34		ND	Neg
					Female	Young	35		ND	Neg
					Male	Young	36		ND	Neg
					Female	Young	37		ND	Pos
C7	5	Schreibers’ bat (*M. schreibersii*)	2015	Cantabria	Female	Young	39	Pos	ND	Neg
					Female	Young	40		ND	Neg
					Female	Young	42		ND	Pos
					Male	Young	43		ND	Pos
					Female	Young	44		ND	Pos
A4	5	Schreibers’ bat (*M. schreibersii*)	2015	Asturias	Male	Young	01A	Pos	ND	Pos
					Male	Young	02A		ND	Neg
					Male	Young	03A		ND	Neg
					Female	Young	04A		ND	Neg
					Male	Young	05A		ND	Neg
A5	5	Schreibers’ bat (*M. schreibersii*)	2015	Asturias	Male	Young	06A	Pos	ND	Pos
					Female	Young	07A		ND	Pos
					Male	Young	08A		ND	Pos
					Male	Young	26B		ND	Neg
					Male	Young	27B		ND	Neg
A1	5	Schreibers’ bat (*M. schreibersii*)	2015	Asturias	Male	Young	01B	Pos	ND	Pos
					Male	Adult	02B		ND	Pos
					Female	Young	04B		ND	Neg
					Male	Adult	06B		ND	Neg
					Female	Adult	08B		ND	Pos
A2	5	Schreibers’ bat (*M. schreibersii*)	2015	Asturias	Male	Adult	10B	Pos	ND	Neg
					Male	Adult	12B		ND	Neg
					Male	Young	13B		ND	Neg
					Female	Young	15B		ND	Neg
					Male	Adult	16B		ND	Neg
A3	5	Schreibers’ bat (*M. schreibersii*)	2015	Asturias	Male	Young	18B	Pos	ND	ND
					Male	Young	22B		ND	ND
					Male	Young	23B		ND	ND
					Male	Young	24B		ND	ND
					Female	Young	25B		ND	ND

**Table 2 viruses-11-00360-t002:** Inferred epitopes detected by Domain Programmable Arrays (DPA) in the Schreiber’s Bent-winged bats samples with evidence of Lloviu (LLOV) exposure. Epitopes are described based on the protein they belong to, amino acid sequence, enrichment value (EXP), position of the first and last amino acid in the sequence (Start and End), and epitopes found in more than one animal (Shared).

**Cantabria Cave**	**Epitopes**
**Sample Number**	**DPA**	**Protein**	**Inferred Epitope**	**EXP**	**Start**	**End**	**Shared**
7	Pos	GP_1_	KPMTDHQEFILQPHS	94	329	343	
8	Pos	GP_2_	ETSSKSAT	>100	28	35	GP2.28
11	Pos	NP	QTQESSDRSDYSRRP	58	511	525	NP.511
		VP40	ASPDKIKS	>100	231	238	VP40.231
14	Pos	GP_2_	LLGSVSNNSSIQELETSSKSAT	>100	14	35	GP2.28
		L	GASFVTDLEKYNLAFRFEFTRPFIEYC	28	622	648	L.620
15	Pos	NP	QTQESSDRSDYSRRP	>100	511	525	NP.511
		NP	GDRPHTTQ	>100	532	539	NP.532
16	Pos	NP	WKQPTSPLSTIPEEE	>100	630	644	NP.630
20	Pos	VP35	DSPQCALIQITKRIPIFGETPP	36	243	271	VP35.243
		GP_2_	LLGSVSNNSSIQELETSSKSAT	33	14	35	GP2.28
27	Pos	VP35	TERTFGKP	49	189	196	VP35.189
		NP	QTQESSDRSDYSRRP	15	511	525	NP.511
		NP	RTLPLISFDDNEGEI	4	567	581	NP.567
37	Pos	VP35	DSPQCALIQITKRIPIFGETPP	5	243	271	VP35.243
		NP	QTQESSDRSDYSRRP	9	511	525	NP.511
		NP	RTLPLISFDDNEGEI	5	567	581	NP.567
42	Pos	GP_2_	ETSSKSAT	32	28	35	GP2.28
43	Pos	VP35	PLIEPKTSANKSTQTENIYQSDQVLREIK	23	42	70	VP35.42
		NP	QTQESSDRSDYSRRP	13	511	525	NP.511
		NP	RTLPLISFDDNEGEI	22	567	581	NP.567
44	Pos	VP35	VREAFDRLEKTEEVTE	7	175	190	VP35.175
		GP_2_	HNATTTSK	>100	98	105	GP2.98
		GP_2_	KTRRRRQVNPVPPTITQQTSTSINTSHHP	>100	105	133	GP2.105
**Asturia Cave**	**Epitopes**
**Sample**	**DPA**	**Protein**	**Inferred Epitope**	**EXP**	**Start**	**End**	
01A	Pos	VP30	NSRITPGDWQCQPCDYPKARFK	12.5	77	98	
		VP35	LEKTEEVTERTFGKP	20	182	196	VP35.189
		NP	RTLPLISFDDNEGEILDDKSD	3	567	583	NP.567
06A	Pos	NP	SQDPNNRQKQSDTQQTQESSDRSDYSRRP	4.2	497	525	NP.511
		NP	RTLPLISFDDNEGEILDDKSDLPAPDTHS	14	567	595	NP.567
		GP_2_	ETSSKSATELTTPINHSQSLQL	9.6	28	49	GP2.28
		VP40	LVPRLMSKDDLGGRDLVMSTKGSCENCYYPGASPTQ	72	287	322	
07A	Pos	GP_1_	TTTLDYDV	20	224	231	GP1.224
		VP35	VREAFDRLE	4.4	175	183	
		NP	LNVDHTIVRKKSIPLFEIGNSDQVCNWIIQIIEAGV	7.1	28	63	
		NP	WKQPTSPLSTIPEEEGGHEANNDNSESDL	76.8	630	658	NP.630
		L	QVLGGLSFLNPEKCF	>100	903	917	
08A	Pos	VP30	YQQHNQES	20	175	182	
		VP35	QSDQVLREIK	13.4	63	70	
		L	AEDIIRPFCEARINLPVQELFKLLPSHYSGNIVHRY	12.9	1232	1267	
01B	Pos	VP24	VKHDLCNFLVTTTITGWDVYWAGHLFHVPNKGIALL	72	196	231	
		GP_1_	TTTLDYDV	>100	224	231	GP1.224
		VP35	VREAFDRLEKTEEVTERTFGKP	6.9	175	196	VP35.189
		NP	STKESSSYTASRTEEDRNNYNS	60	441	462	NP.441
		NP	RTLPLISFDDNEGEILDDKSDLPAPDTHS	8.8	567	595	NP.567
		NP	WKQPTSPLS	4.2	630	638	NP.630
		L	ANTVMTSLLADMNNA	19.6	1421	1435	
02B	Pos	VP35	LEKTEEVTERTFGKP	3.1	182	196	VP35.189
		NP	NVDHITDLLGVGSRDKSLRKTLSALEFEP	12.5	112	140	
08B	Pos	VP35	VREAFDRLEKTEEVTERTFGKP	36	175	196	VP35.189
		NP	STKESSSYTASRTEEDRNNYNS	>100	441	462	
		GP_2_	ETSSKSAT	57	28	35	GP2.28
		GP_2_	LASVTNTPTP	28.7	49	56	
		L	DHVQVRGASFVTDLEK	53.7	616	637

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
