# Peer review of "First Evidence of Antibodies Against Lloviu Virus in Schreiber’s Bent-Winged Insectivorous Bats Demonstrate a Wide Circulation of the Virus in Spain"

_viruses, 2019, doi:10.3390/v11040360_

Round 1
Reviewer 1 Report
The manuscript by Dr. Eva Ramírez de Arellano and colleagues presents the results of a multi-method based serologic survey for Lloviu cuevavirus in Spanish bat and human sera. Considering the significant lack of knowledge about Lloviu virus biology, and its recent report from another part of Europe, the manuscript represents a great interest for researchers in this field. I am pleased to have the chance of revising this work.
I believe the manuscript fits the scope of the journal and can be published, however, I have some major comments on the basic concept, research planning and methods.
Comments in details:
What is the logic behind sampling Miniopterus schreibersii (Msch) bats from the LLOV affected cave and compare this data with other species (Eptesicus serotinus) from a totally different and presumably unaffected cave? In addition, these two species do not regularly co-roost, it would be more useful and logical to compare with Myotis myotis or Rinolophus species which are more sympatric than Serotines. Please make it clear in the manuscript why the authors surveyed these specific sites or include the limitations in the discussion part.
The authors used pooled samples, which enhances the logistics behind massive surveillance studies but limits the specificity and the possibility to gain detailed results. It is perfect for pre-screening but it is insufficient to draw deep conclusions. I understand the limitation of antigen amount used as it is indicated in Lines 197-198, but maybe it would be great to include more tests if possible, by getting more antigens.
In some parts of the manuscript authors should tone down the conclusions regarding the connection between Llov and bat mortality (line 252) - it is not proven yet just suspected.
In contrast they conclude the possible independence between mortalities and infections based on the screening of feces samples. Based on literature data, Llov RNA was detected from tissue samples mainly and only rectal swabs were PCR positive from Cantabria. PCR positivity was describes for only lung and spleen from Hungarian cases.
The authors provided some important details regarding epitope prediction. I cannot find the methodical background in methods sections. Unfortunately the authors only provided details about epitope prediction in the discussion part and the made? a figure as a supplement in connection with this. I really miss the integration of this part in the whole manuscript.
Minor comments in details:
Line 33: Why these bats were considered as potentially exposed? Is this the same colony which was affected in 2002-2003? Is there any data from local chiropterologist which supports the identity of this colony from 2015 (eg: ringing data or other). If yes please indicate this important clue in the MS. Is it possible that the original colony was dispersed or perished and other sub-population filled the roost site? - there are several examples for this and it would include some more discussion in the manuscript - where they were affected by the virus.
Line 40: Finish the sentence with a dot.
Line 43: Based on my experience it is really hard to measure the health of bats, i would not consider them healthy just from visual signs, especially Miniopterus bats, please tone down this conclusion in the end of the abstract because based on just this observation it is hard to dissociate.
Introduction:
Line 49: Please change genus to genera.
Line 51: The factors which triggered the emergence of LLOV in 2002 are still unclear, along with its appearence in Hungary. It is also a possibility that it was imported from other territories, we still do not know. The die-off in Spain and the mortality in Hungary is normally unusual, and its detection along with LLOV may suggest some connection. It is still hard to tell anything exact.
Materials and methods
Line 92: Refer to the major comment about Serotines
Table 1: I have an idea about this table, it would be practical and much more informative if the authors would compress it to a Map with the localities, positives, negatives and other data. Data which dont fit the map can be included in the map description.
Line 112: It can be a bug in the system but i could not find the supplement text, just the picture of blotting
Line 120: Please include the ful name of Dr Takada
Line 114-117: The authors refer to the conditions of qPCR to the original article of Negredo and colleagues. Unfortunately, in that manuscript it is not included and can be only requested, which makes this reference and section to a dead-end for readers. Please make it clear if the authors want to publish the primers and test or not, it fits the word limit.
Line 146: If the samples were processed in triplicates, why do we have cotradiction in the table, where there is for example a positive immunoblot pool with negative individual samples (e.g.: A2). Please make it clear how is this possible.
Line 160-162: I would not conclude the oddity of die-off episodes based on just feces screening. As I mentioned earlier it would be better to include more non-invasive samples for PCR screening of even tissue samples from dead animals and make stonger baseline for conclusions.
Please improve the quality of Supplementary figure 1 A, B, and C
Line 187: Why these animals were sampled in a distant region. Please do not draw conclusions for LLOV if the territory was presumably not affected, but refer to these samples as controls.
Line 223-224: It is hard to make exact data on seroprevalence from low sample numbers, it can be misleading in the future.
Line 252: Llov was only suspected to be connected with the mortality, in other parts of the MS the authors are on different opinion. Please tone down this sentence and indicate the lack of knowledge about this connection.
Line 267: What if die-off events remained simply undetected. Msch populations are declining all across Europe, mainly because of human disturbance. Tha majority of their summer or winter roosting sites are unknown or if known they are protected habitats and rarely reached by specialists. It can be an explanation for undetected cases between 2002 and 2016.
Line 271-291: Please integrate this part to MatMethods and Results part, and leave just the discussion here.
Line 305: In Hungarian positive animals, viral load was presumably low as the publication
316: It may be highly pathogenic or not, or this study can be the first proof of immunity in survived animals after Llov enzootics. Many scenarios are possible.
Author Response
What is the logic behind sampling Miniopterus schreibersii (Msch) bats from the LLOV affected cave and compare this data with other species (Eptesicus serotinus) from a totally different and presumably unaffected cave? In addition, these two species do not regularly co-roost, it would be more useful and logical to compare with Myotis myotis or Rinolophus species which are more sympatric than Serotines. Please make it clear in the manuscript why the authors surveyed these specific sites or include the limitations in the discussion part.
1.1) We thank the reviewer for pointing out this issue. The rationale of using serum samples from unaffected populations of a bat species with a completely different ecology is to use them as negative controls in order to identify possible unspecific results. E. serotinus was specifically chosen for this reason. The search of LLOV antibodies in species co-habiting with M.schreibersii, as well as in M. schreibersii from other locations not affected by the 2002-2003 mass mortality would be matter of future studies. We have clarified this issue in the abstract (line 34, introduction (lines 87 and 103), results (lines 249-250).
The authors used pooled samples, which enhances the logistics behind massive surveillance studies but limits the specificity and the possibility to gain detailed results. It is perfect for pre-screening but it is insufficient to draw deep conclusions. I understand the limitation of antigen amount used as it is indicated in Lines 197-198, but maybe it would be great to include more tests if possible, by getting more antigens.
1.2) We acknowledge the reviewer comment about the restrictions of the study due the antigen limitations. The aim of this prospective study was to mainly check whether anti-LLOV antibodies were or not present in M. schreibersii populations in the wild. The results showed in this manuscript attest to this fact and invite to follow the research with more extensive and detailed studies. We first did a WB screening with pools to screen for the presence of anti-LLOV antibodies. After that initial screening, we focused on the characterization of the antibody reactivity by DPA where the samples were tested individuality. Due to the grant conclusion, there is currently not possibility to produce more antigen.
In some parts of the manuscript authors should tone down the conclusions regarding the connection between Llov and bat mortality (line 252) - it is not proven yet just suspected.
1.3) We have tone down our initial statements regarding bat mortality and LLOV infection, in agreement with the reviewer (lines 320-321).
In contrast they conclude the possible independence between mortalities and infections based on the screening of feces samples. Based on literature data, Llov RNA was detected from tissue samples mainly and only rectal swabs were PCR positive from Cantabria. PCR positivity was describes for only lung and spleen from Hungarian cases.
1.4) We agree with the reviewer, and we have changed in the manuscript to address this issue (lines 222-223).
The authors provided some important details regarding epitope prediction. I cannot find the methodical background in methods sections. Unfortunately the authors only provided details about epitope prediction in the discussion part and the made? a figure as a supplement in connection with this. I really miss the integration of this part in the whole manuscript.
1.5) We thank the reviewer for pointing out this mistake. To address this omission, we have expanded the methods section on “Detection of LLOV-specific Antibodies by Domain Programmable Arrays” (lines 195-202) and added a “Protein modelling and surface epitope visualization” section (lines 204-212) to give more clarity.
Minor comments in details:
Line 33: Why these bats were considered as potentially exposed? Is this the same colony which was affected in 2002-2003? Is there any data from local chiropterologist which supports the identity of this colony from 2015 (eg: ringing data or other). If yes please indicate this important clue in the MS. Is it possible that the original colony was dispersed or perished and other sub-population filled the roost site? - there are several examples for this and it would include some more discussion in
1.6) We agree with the reviewer. SBWB were captured from the same cave where Lloviu virus was discovered but we do not know if those bats are the same colony. Most likely, the original colony has dispersed or perished and other sub-population filled the roost site, as the reviewer said. We have included a sentence to discuss this point (paragraph at 330-332).
Line 40: Finish the sentence with a dot.
1.7) We have added the dot to the sentence.
Line 43: Based on my experience it is really hard to measure the health of bats, i would not consider them healthy just from visual signs, especially Miniopterus bats, please tone down this conclusion in the end of the abstract because based on just this observation it is hard to dissociate.
1.8) We thank to reviewer to point out this mistake. We have replaced “healthy” for “live captured” throughout the manuscript, to stress these animal were alive, not dead, when captured.
Introduction:
Line 49: Please change genus to genera.
1.9) We have changed genus to genera to address the reviewer.
Line 51: The factors which triggered the emergence of LLOV in 2002 are still unclear, along with its appearence in Hungary. It is also a possibility that it was imported from other territories, we still do not know. The die-off in Spain and the mortality in Hungary is normally unusual, and its detection along with LLOV may suggest some connection. It is still hard to tell anything exact.
1.10) We modified the sentence to address the reviewer concerns (first paragraph of the introduction)
Materials and methods
Line 92: Refer to the major comment about Serotines
1.11) Please, see response 1.1).
Table 1: I have an idea about this table, it would be practical and much more informative if the authors would compress it to a Map with the localities, positives, negatives and other data. Data which dont fit the map can be included in the map description.
1.12) We thank the reviewer for this idea. We decided to make a table that included the complete information of all samples tasted. Our attempt to make a map turned the figure convoluted and hard to read.
Line 112: It can be a bug in the system but i could not find the supplement text, just the picture of blotting.
1.13) We apologize for this mistake. We will included the supplementary text in the rebuttal submission.
Line 120: Please include the ful name of Dr Takada
1.14) We added the full name of Dr. Ayato Takada (line 403), and his initial (A.) on lines 156 and 172.
Line 114-117: The authors refer to the conditions of qPCR to the original article of Negredo and colleagues. Unfortunately, in that manuscript it is not included and can be only requested, which makes this reference and section to a dead-end for readers. Please make it clear if the authors want to publish the primers and test or not, it fits the word limit.
1.15) We thank the reviewer for pointing out this omission. We included the conditions in the manuscript (lines 140-153)
Line 146: If the samples were processed in triplicates, why do we have cotradiction in the table, where there is for example a positive immunoblot pool with negative individual samples (e.g.: A2). Please make it clear how is this possible.
1.16) Although generally in agreement, some discordances were observed between results from the immunoblot and DPA. Only 8 individual samples were analysed both by DPA and immunoblot. At the pool level, two pools (pools C1 and A2) out of the total of 12 generated also resulted in discordant results (Table 1). The discrepancy in these results is likely due to the higher immunoblot sensitivity compared with DPA, since we are utilizing a very conservative (high) threshold in the enrichment factor for DPA (Discussion: lines 339-341). An alternate explanation would be to consider that DPA would be more accurate in detecting to LLOV-specific antibodies than immunoblots, or a combination of both. Nonetheless, to estimate the seroprevalence of LLOV in SBWB, we only considered the results where both assays correlate.
Line 160-162: I would not conclude the oddity of die-off episodes based on just feces screening. As I mentioned earlier it would be better to include more non-invasive samples for PCR screening of even tissue samples from dead animals and make stonger baseline for conclusions.
1.17) We made changes and added a sentence in the manuscript to explain better this conclusion according to the reviewer comment
Please improve the quality of Supplementary figure 1 A, B, and C
1.18) Following the reviewer recommendations, we have improved the quality of Supplementary figure 1 A, B, and C (this figure is now Figure 1).
Line 187: Why these animals were sampled in a distant region. Please do not draw conclusions for LLOV if the territory was presumably not affected, but refer to these samples as controls.
1.19) Please, see response 1.1)
Line 223-224: It is hard to make exact data on seroprevalence from low sample numbers, it can be misleading in the future.
1.20) We added a sentence to explain this part (lines 268-270).
Line 252: Llov was only suspected to be connected with the mortality, in other parts of the MS the authors are on different opinion. Please tone down this sentence and indicate the lack of knowledge about this connection.
1.21) We tone down this sentence (lines 320-321)
Line 267: What if die-off events remained simply undetected. Msch populations are declining all across Europe, mainly because of human disturbance. Tha majority of their summer or winter roosting sites are unknown or if known they are protected habitats and rarely reached by specialists. It can be an explanation for undetected cases between 2002 and 2016.
1.22) We introduced this explanation in the discussion part of the manuscript (line 330-332)
Line 271-291: Please integrate this part to Mat Methods and Results part, and leave just the discussion here.
1.23) We thank the reviewer suggestion, however this portion is essential for the discussion.
Line 305: In Hungarian positive animals, viral load was presentencesumably low as the publication
1.24) We removed this publication refereeing this sentence.
Line 316: It may be highly pathogenic or not, or this study can be the first proof of immunity in survived animals after Llov enzootics. Many scenarios are possible.
1.25) We expanded the discussion to address the reviewer concerns (lines 381-388).
Reviewer 2 Report
In 2002, RNA genomes of Lloviu virus (LLOV) was first detected in the lungs, livers, rectal swabs, and/or spleen of Schreiber's Bent-winged bat (SBWB). However, the ecology of LLOV remains unclear. In this manuscript, the authors attempted to detect the specific antibody against LLOV from SBWB by conducting a serosurvey for Lloviu virus (LLOV) in two bat species (SBWB and Serotine bat) and human. They found the presence of the specific antibodies in SBWB without any symptoms. This finding might provide important and valuable information for the understanding of the ecology of LLOV. However, there are several important points that need to be clarified and modified in this manuscript. Moreover, the overall quality (English grammar, presentation of the data, etc.) should be largely improved. Following are points that may improve the manuscript.
Major comments
1. Throughout the manuscript, there are so many typographical and grammatical errors. The authors should consider asking someone (such as English native speakers and a professional editing service) to proofread the manuscript.
2. tivated, the conditions should be added in the “Materials and Methods” section (e.g., All sera were inactivated at 56°C for 30 minutes prior to further analyses.)
3. Lines 104, and 257-259: Why did the authors use only GP2 as an antigen for the immunoblot assays? It seems that full-length GP is more suitable as the antigen.
4. Line 108: GP2 has a transmembrane domain (TM) and a cytoplasmic tail (CT) at the C-terminus. Although “Supplementary Figure 1B” showed that GP2 was detected in the supernatants, did the 6xHis-LLOV-GP2 protein have also the TM or the CT?
5. Line 112: “Supplementary text” is missing.
6. Line 116: More explanation and description for the methodology of qRT-PCR are needed. Reference paper by Negredo et al [1] did not provide technical information, such as the name of used kits and PCR conditions.
7. Lines 126-137: What kind of blocking solution was used?
8. Lines 146-147: Why did the authors use a hyperimmune polyclonal serum against EBOV? Present study attempted to investigate the prevalence rate of LLOV in SBWB. It seems that an anti-LLOV GP polyclonal mouse serum (line 120) is more suitable as a positive control.
9. Lines 168-169: Did the “GenBank AN: JF828358” be used for production of GP1? If so, the authors should attempt to express and purify LLOV GP1 again. LLOV GP has an RNA editing site in its GP sequence. “JF828358” has only the “7A”; however, the full-length GP need to have “8A”. Thus, the authors must modify the GP1 sequence by themselves. The failure of GP1 production might be caused by the incorrect GP1 sequence.
10. Line 206: What is the “enrichment value (EXP)”?
11. Lines 214-218: The cause of discordant results should be discussed in the “Discussion” section.
12. Line 220: Although the authors described “Group 2 comprised 60 serum samples from SBWB” (line 89), only 52 serum samples were analyzed. Why?
13. Lines 267-268: Indeed, the results of this manuscript indicate that some SBWB were infected with LLOV in the past; however, the results don’t provide the evidence that SBWB were frequently exposed to LLOV.
14. Lines 268-271: Ref 34 focused on only the ERBs, while this study focused on SBWBs. It is not reasonable that immune systems in bats are similar just because of the chiropteran animal. Moreover, it remains a possibility that a neutralizing antibody in SBWB can clear LLOV. Thus, the findings in this manuscript cannot exclude the performing neutralization tests.
15. It is hard to understand what Supplementary figure 3 shows. The authors should add more explanations for methods (e.g. how to generate structure models, how to overlay the structures, monomer or dimer, PDB ID, software, and so on).
Minor comments
1. Title: LLOV is not a general term for readers. Spell out LLOV.
2. Line 66: According to the latest ICTV Taxonomy, Bombali virus (BOMV) has not yet been assigned to genus Ebolavirus. “Ebolavirus” should be replaced with “filovirus”.
3. Table 1: The authors used “+” or “-” for the result of the pool immunoblot. Whereas “pos” or “neg” for the result of the individual immune blot and DPA were used. These terms should be unified.
4. Line 120, “an anti-LLOV GP polyclonal mouse serum”: Suitable reference should be added.
5. Line 136: Abbreviations for CAT should be defined somewhere in the text.
6. Line 140: “every strain of filovirus” should not be used. EBOV, SUDV, TAFV, BDBV, RESTV, MARV, and RAVV are virus name but not strain name.
7. Lines 140-141: Abbreviations for EBOV, SUDV, TAFV, BDBV, RESTV, MARV, RAVN should be defined somewhere in the text. And “RAVN” should be “RAVV”.
8. Supplementary Figure 1A and 1C: What did the authors intend to put these figures? There was no explanation in the main text.
9. Line 395: Article title is incorrect. “Reverse-Transcription Polymerase Chain Reaction Kit” should be “Diagnostic Reverse-Transcription Polymerase Chain Reaction Kit”.
10. Supplementary figure 2: “Nuc” should be “NP”; “OLIGO” should be “Oligo peptide”; SUD should be SUDV; The label of the “OLIGO” was too complicated.
Author Response
Major comments
1. Throughout the manuscript, there are so many typographical and grammatical errors. The authors should consider asking someone (such as English native speakers and a professional editing service) to proofread the manuscript.
2.1) We addressed this issue.
2. Lines 84-97: More explanation about serum samples tested in this study should be addressed (i.e. inactivation or not). If serum samples were inactivated, the conditions should be added in the “Materials and Methods” section (e.g., All sera were inactivated at 56°C for 30 minutes prior to further analyses.)
2.2) To address the reviewer comment, we added more explanation in this point about the serum samples. The serum samples were not inactivated (lines 107-112).
3. Lines 104, and 257-259: Why did the authors use only GP2 as an antigen for the immunoblot assays? It seems that full-length GP is more suitable as the antigen.
2.3) 2.3) We understand the reviewer concern. Purification of the full-length GP was not successful because the protein was retained in the column resin. So, we did the immunoblots experiments only with the antigen GP2.
4. Line 108: GP2 has a transmembrane domain (TM) and a cytoplasmic tail (CT) at the C-terminus. Although “Supplementary Figure 1B” showed that GP2 was detected in the supernatants, did the 6xHis-LLOV-GP2 protein have also the TM or the CT?
2.4) Yes, the 963 bp amplified GP2 complete fragment was directionally subcloned to construct a recombinant vector pfastbac-HIS-LLOV-GP2 with a hexahistidine (6xHis) tag sequence.
5. Line 112: “Supplementary text” is missing.
2.5) Please, see response 1.13)
6. Line 116: More explanation and description for the methodology of qRT-PCR are needed. Reference paper by Negredo et al [1] did not provide technical information, such as the name of used kits and PCR conditions.
2.6) Please, see response 1.15)
7. Lines 126-137: What kind of blocking solution was used?
2.7) We thank the reviewer to point out this omission. We added the blocking solution in lines 169-170.
8. Lines 146-147: Why did the authors use a hyperimmune polyclonal serum against EBOV? Present study attempted to investigate the prevalence rate of LLOV in SBWB. It seems that an anti-LLOV GP polyclonal mouse serum (line 120) is more suitable as a positive control.
2.8) The hyperimmune polyclonal serum against EBOV was used as a positive control (Material and methods, lines 187-188) for the assay. DPA contains oligonucleotides for all the filoviruses and this control enriches EBOV specific clusters, what it is a better control than a LLOV anti-serum, to ensure the specificity of the assay.
9. Lines 168-169: Did the “GenBank AN: JF828358” be used for production of GP1? If so, the authors should attempt to express and purify LLOV GP1 again. LLOV GP has an RNA editing site in its GP sequence. “JF828358” has only the “7A”; however, the full-length GP need to have “8A”. Thus, the authors must modify the GP1 sequence by themselves. The failure of GP1 production might be caused by the incorrect GP1 sequence.
2.9) We thank the review for pointing out this potential issue. The production problem was not related to the editing site, as this was corrected to be in frame. The problem to produce the recombinant GP1 protein was the retention of the protein in the column resin.
10. Line 206: What is the “enrichment value (EXP)”?
2.10) We thank the reviewer for pointing out this omission. We have added a sentence to explain this (lines 201-202)
11. Lines 214-218: The cause of discordant results should be discussed in the “Discussion” section.
2.11) The discordant results between the immunoblot and DPA are discussed in 333-349. Please, see also response 1.16).
12. Line 220: Although the authors described “Group 2 comprised 60 serum samples from SBWB” (line 89), only 52 serum samples were analyzed. Why?
2.12) We thank the reviewer for this concern. Unfortunately, after doing the immunoblot initial screening, we run out of serum from the missing samples to be also analysed by DPA.
13. Lines 267-268: Indeed, the results of this manuscript indicate that some SBWB were infected with LLOV in the past; however, the results don’t provide the evidence that SBWB were frequently exposed to LLOV.
2.13) We agree with the reviewer in this point. We change this sentence in discussion (line 344).
14. Lines 268-271: Ref 34 focused on only the ERBs, while this study focused on SBWBs. It is not reasonable that immune systems in bats are similar just because of the chiropteran animal. Moreover, it remains a possibility that a neutralizing antibody in SBWB can clear LLOV. Thus, the findings in this manuscript cannot exclude the performing neutralization tests.
2.14) We thank the reviewer for spotting this point. We have adequate the text to address this (line 346 -351)
15. It is hard to understand what Supplementary figure 3 shows. The authors should add more explanations for methods (e.g. how to generate structure models, how to overlay the structures, monomer or dimer, PDB ID, software, and so on).
2.15) To address the reviewer concern, we made Supplementary figure 3 a main figure (Figure 3) and added methods lines 196-202.
Minor comments
1. Title: LLOV is not a general term for readers. Spell out LLOV.
2.2.1) We thank the reviewer for pointing out this mistake. LLOV was spelled out in the title.
2. Line 66: According to the latest ICTV Taxonomy, Bombali virus (BOMV) has not yet been assigned to genus Ebolavirus. “Ebolavirus” should be replaced with “filovirus”.
2.2.2) We thank the reviewer for pointing out this. We corrected the error.
3. Table 1: The authors used “+” or “-” for the result of the pool immunoblot. Whereas “pos” or “neg” for the result of the individual immune blot and DPA were used. These terms should be unified.
2.2.3) We thank the reviewer for finding this incongruence. We fixed table 1.
4. Line 120, “an anti-LLOV GP polyclonal mouse serum”: Suitable reference should be added.
2.2.4) We thank the reviewer for pointing out this omission. We added the suitable reference.
5. Line 136: Abbreviations for CAT should be defined somewhere in the text.
2.2.5) We thank the reviewer for pointing out this omission. We added the definition of CAT.
6. Line 140: “every strain of filovirus” should not be used. EBOV, SUDV, TAFV, BDBV, RESTV, MARV, and RAVV are virus name but not strain name.
2.2.6) We thank the reviewer for pointing this out. We have modified this part to better explain that DPA contains every single available sequence at GenBank, not only one representative sequence of the virus (lines 178-182).
7. Lines 140-141: Abbreviations for EBOV, SUDV, TAFV, BDBV, RESTV, MARV, RAVN should be defined somewhere in the text. And “RAVN” should be “RAVV”.
2.2.7) We thank the reviewer for pointing out this mistake. We corrected this error.
8. Supplementary Figure 1A and 1C: What did the authors intend to put these figures? There was no explanation in the main text.
2.2.8) The explanation of the figure 1A is mentioned in the supplementary text and 1c have been added in results: “Presence of antibodies against LLOV GP2 protein in SBWB by Immunoblot”
9. Line 395: Article title is incorrect. “Reverse-Transcription Polymerase Chain Reaction Kit” should be “Diagnostic Reverse-Transcription Polymerase Chain Reaction Kit”.
2.2.9) We thank the reviewer for pointing out this inaccuracy. We have corrected it.
10. Supplementary figure 2: “Nuc” should be “NP”; “OLIGO” should be “Oligo peptide”; SUD should be SUDV; The label of the “OLIGO” was too complicated.
2.2.10) We thank the reviewer for pointing out this inconsistency. The naming of the oligo nucleotides was imported from GenBank. It retains the denomination in this database. We explained this in the figure legend.
Reviewer 3 Report
de Arellano and colleagues investigated the presence of antibodies against Lloviu virus in Schreiber's bent-winged insectivorous bats from Spain. For this, they employed immunoblot and domain programmable arrays (DPA). They detected antibodies reactive against LLOV proteins in some Schreibers bats but not Serotine bats or humans. Information on the frequency of LLOV infection of bats and zoonotic transmission is of high interest. However, several important issues remain to be addressed:
Major
It is essential to know whether and how the recombinant GP2 was purified. For this, the authors refer the reader to the supplementary material. However, the supplement only contains a figure identical to figure S1 in the manuscript.
Supplementary figure 1A and 1B seem to show immunoblots that confirm expression of GP2 and CAT control protein. The two proteins plus the non-related protein – please specify what has been used – should be loaded on one single gel and analyzed. In addition, the results of Coomassie staining should be presented so that the reader can judge protein purity.
The authors should show the immunoblots for all pools of Schreibers bat and Serotine bat serum samples as well as for the individual sera that were analyzed by DPA. This is important since it allows the reader to judge the quality of the analysis.
In the view of this reviewer, it is important to include several negative sera in the DPA analysis which should not result in the enrichment of certain sequences.
Only 8 single sera were analyzed by both immunoblot and DPA and for 2 sera discordant results were obtained. Similarly, pool A2 was positive by immunoblot but all single sera were negative by DPA. How reliable is the DPA assay?
Minor
The text requires revision for occasional grammar, style and formatting issues.
The authors should cite PMID: 30617348
Detection of antibodies does not allow concluding that “LLOV is still circulating among SBWB”
The supplementary figures should be regular figures.
Author Response
It is essential to know whether and how the recombinant GP2 was purified. For this, the authors refer the reader to the supplementary material. However, the supplement only contains a figure identical to figure S1 in the manuscript.
3.1) We thank the reviewer for pointing out his omission. We included the supplementary text
Supplementary figure 1A and 1B seem to show immunoblots that confirm expression of GP2 and CAT control protein. The two proteins plus the non-related protein – please specify what has been used – should be loaded on one single gel and analyzed. In addition, the results of Coomassie staining should be presented so that the reader can judge protein purity.
3.2) The conditions are specified in the supplementary text attached. The two proteins were loaded and analysed in the same gel.
The authors should show the immunoblots for all pools of Schreibers bat and Serotine bat serum samples as well as for the individual sera that were analyzed by DPA. This is important since it allows the reader to judge the quality of the analysis.
3.3) We thanks the reviewer for this critical comment. We decided not to include the pictures of all pools and individuals sera analysed because of the large number of immunoblots, since each pool or individual sera were tested in different immunoblot with a positive and a negative controls. We included just an example in the figure 1c
In the view of this reviewer, it is important to include several negative sera in the DPA analysis which should not result in the enrichment of certain sequences.
3.4) We thank the reviewer for this comment. DPA is run with several negative and positive controls. We did not include the negative because it would be an empty plot. We set the condition of the assay to minimize enrichment in the negative samples.
Only 8 single sera were analyzed by both immunoblot and DPA and for 2 sera discordant results were obtained. Similarly, pool A2 was positive by immunoblot but all single sera were negative by DPA. How reliable is the DPA assay?
3.5) Please, see response 1.16)
Minor
The text requires revision for occasional grammar, style and formatting issues.
3.6) Please, see response 2.1)
The authors should cite PMID: 30617348
3.7) We thank the reviewer for pointing out this omission. This reference has been cited
Detection of antibodies does not allow concluding that “LLOV is still circulating among SBWB”
3.8) We thank the reviewer for pointing out this wrong assumption. We changed this affirmation in the text.
The supplementary figures should be regular figures.
3.9) We decided to follow the reviewer suggestion, and now we changed supplementary figures as figure.
Round 2
Reviewer 1 Report
I would like to thank the authors for the great improvement made in the manuscript. In my opinion, it is ready for publication in the journal. I would also thank the opportunity to revise this work.
All reviewer points were answered correctly.
I only suggest performing a final spell checking before acceptance (e.g see line 87 and correct human to humans).
Author Response
I would like to thank the authors for the great improvement made in the manuscript. In my opinion, it is ready for publication in the journal. I would also thank the opportunity to revise this work.
All reviewer points were answered correctly.
I only suggest performing a final spell checking before acceptance (e.g see line 87 and correct human to humans).
We would like to thank the reviewer for considering our manuscript almost ready to be published. We performed a final read-through to correct grammatical errors.
Reviewer 2 Report
Although the manuscript has been improved, there are still some important issues to be addressed.
Major comments
1. Lines 226-228: An explanation on LLOV GP1 and GP2 subunits should be added in the Introduction.
2. The quality of Figure 3 is too low.
1) Lines 204-212, Figure 3: It doesn’t seem that the predicted structures of LLOV proteins are good models. Judging from the sequence identity for each protein of EBOV and LLOV, the structures must be more similar between EBOV and LLOV. Several kinds of homology modeling server, such as SWISS-MODEL, can be utilized.
2) Lines 353-357, Figure 3: It is hard for readers to understand the locations of “LLOV NP.511-525, NP567-581 and NP.630-638” or “EBOV NP 521-540, 561-580, and 632-645” in Figure 3 as well as LLOV epitopes. The authors should add labels in Figure 3.
3) What does “orange” show in the structural model of EBOV?
4) What is the numbering system of these epitopes, EBOV numbering or LLOV numbering?
Minor comments
1. There are still many typographical and grammatical errors left on the current manuscript. The authors should check carefully through the manuscripts, and should correct them. For example,
1) Line 114: Replace “-20C” with “-20°C”.
2) Line 115: Replace “-80C” with “-80°C”.
3) Line 361: Replace “RESV” with “RESTV”.
4) Line 209: Replace “AER23672.1_VP35” with “AER23672.1”.
5) Table 2: “VP-40”, “VP-35”, and “GP2” should be replaced with “VP40”, “VP35”, and “GP2”, respectively.
Author Response
Major comments
1. Lines 226-228: An explanation on LLOV GP1 and GP2 subunits should be added in the Introduction.
To address the reviewer’s concerns, we added a paragraph in the introduction (lines 83-94) to explain this point about LLOV GP1 and GP2 subunits.
2. The quality of Figure 3 is too low.
Figure 3 was generated to meet print requirements.
1) Lines 204-212, Figure 3: It doesn’t seem that the predicted structures of LLOV proteins are good models. Judging from the sequence identity for each protein of EBOV and LLOV, the structures must be more similar between EBOV and LLOV. Several kinds of homology modeling server, such as SWISS-MODEL, can be utilized.
We thank the reviewer for their critiques. The international blind trials of protein structure prediction methods found no significant difference in accuracy of the top web servers for protein modelling, including Phyre2, i-TASSER, Swiss-Model, HHpred, PSI-Pred, Roetta, and Raptor [PMC5298202]. Phyre2 performed as well as other programs at modelling proteins with<30% homology="" overall="" to="" a="" known="" structure.="" we="" tested="" i-tasser="" and="" phyre2="" with="" both="" llov="" ebov="" found="" no="" differences.="" selected="" based="" on="" ease="" of="" use="" high="" prediction="" scores.="" submitted="" ebola="" zaire="" sequences="" compare="" as="" control="" the="" structures="" for="" sudan="" received="" similar="" scores="" predictive="" percentage="" at="">90% confidence for Ebola Zaire at similar rates as what was reported for LLOV. The only residues modelled at low confidence are part of the mucin-like domain, as in other filovirus models since it cannot be structurally modelled.
2) Lines 353-357, Figure 3: It is hard for readers to understand the locations of “LLOV NP.511-525, NP567-581 and NP.630-638” or “EBOV NP 521-540, 561-580, and 632-645” in Figure 3 as well as LLOV epitopes. The authors should add labels in Figure 3.
Labels were added to figure 3
3) What does “orange” show in the structural model of EBOV?
The orange shows a region in each protein with known epitopes similar to the regions defined in LLOV. We added this to the figure legend.
4) What is the numbering system of these epitopes, EBOV numbering or LLOV numbering?
The numbering system corresponds to each specific species proteins, EBOV numbering for EBOV proteins, and LLOV numbering for LLOV proteins.
Minor comments
1. There are still many typographical and grammatical errors left on the current manuscript. The authors should check carefully through the manuscripts, and should correct them. For example,
1) Line 114: Replace “-20C” with “-20°C”.
2) Line 115: Replace “-80C” with “-80°C”.
3) Line 361: Replace “RESV” with “RESTV”.
4) Line 209: Replace “AER23672.1_VP35” with “AER23672.1”.
5) Table 2: “VP-40”, “VP-35”, and “GP2” should be replaced with “VP40”, “VP35”, and “GP2”, respectively.
We addressed all these issues raised by the reviewers and we checked carefully the entire manuscript.
Reviewer 3 Report
The following major points remain to be addressed:
Supplementary figure 1A and 1B seem to show immunoblots that confirm expression of GP2 and CAT control protein. The two proteins plus the non-related protein – please specify what has been used – should be loaded on one single gel and analyzed. In addition, the results of Coomassie staining should be presented so that the reader can judge protein purity.
Figure 1A must show expression of CAT (the control protein) and LLOV-GP2 in supernatants and pellets of insect cells on on
e single gel. However, LLOV-GP2 expression is not shown. Therefore, a revised figure 1A must be presented. The authors can additionally show expression of a second control protein but then the identity of the control protein must be stated. In addition, figure 1A must be referenced in the manuscript text.
If an attempt was made to purify LLOV-GP2 using for instance the HIS tag then it is essential to show Coomassie or sliver staining of the purified protein and to document the methods used. If LLOV-GP2 was not purified then it should be clearly stated that crude, unpurified cell lysates (or culture supernatants) were used throughout the study and not purified protein.
The authors should show the immunoblots for all pools of Schreibers bat and Serotine bat serum samples as well as for the individual sera that were analyzed by DPA. This is important since it allows the reader to judge the quality of the analysis.
Showing more unprocessed data is essential to judge the quality of the study. Viruses does not impose restrictions on article length, the counterarguments raised by the authors are therefore not valid.
Author Response
Supplementary figure 1A and 1B seem to show immunoblots that confirm expression of GP2 and CAT control protein. The two proteins plus the non-related protein – please specify what has been used – should be loaded on one single gel and analyzed. In addition, the results of Coomassie staining should be presented so that the reader can judge protein purity.
Figure 1A must show expression of CAT (the control protein) and LLOV-GP2 in supernatants and pellets of insect cells on one single gel. However, LLOV-GP2 expression is not shown. Therefore, a revised figure 1A must be presented. The authors can additionally show expression of a second control protein but then the identity of the control protein must be stated. In addition, figure 1A must be referenced in the manuscript text
CAT (the control protein) and LLOV-GP2, in supernatants and pellets of insect cells, were loaded and analysed in different gels. Figure 1A shown the expression of CAT and figure 1B shown the expression of GP2. We understand the reviewer concern, unfortunately there is no more sample available to perform the requested combined gel. In addition, figure 1A have been referenced in the manuscript text (line 141).
If an attempt was made to purify LLOV-GP2 using for instance the HIS tag then it is essential to show Coomassie or sliver staining of the purified protein and to document the methods used. If LLOV-GP2 was not purified then it should be clearly stated that crude, unpurified cell lysates (or culture supernatants) were used throughout the study and not purified protein.
We have specified this in lines 145-149 of methods: “The 40 kDa recombinant 6xHis-LLOV-GP2 protein used as antigen was obtained from a crude extract of the pellet fraction after treatment with Inclusion Body Solubilization reagent (IBS, Thermo Fisher scientific). A detailed summary of the antigen production process is included in a supplementary text (see supplementary data)”. In addition, in Figure 1, we referred LLOV-GP2 as: “Expression of the recombinant LLOV GP2 protein (40 KDa) in the supernatant (lane 1) and pellet (lane 2) of the crude extracted lysate….”.
The authors should show the immunoblots for all pools of Schreibers bat and Serotine bat serum samples as well as for the individual sera that were analyzed by DPA. This is important since it allows the reader to judge the quality of the analysis.
Showing more unprocessed data is essential to judge the quality of the study. Viruses does not impose restrictions on article length, the counterarguments raised by the authors are therefore not valid.
We understand the reviewer’s concern, unfortunately the pictures generated during this study have a diagnostic value but not a publishable quality. The lack of more LLOV GP2 as an antigen for the immunoblot precludes us to perform a new gel to obtain a better quality picture. Below are the original pictures for your evaluation.
